# L-Fucose Suppresses Lipid Accumulation via the AMPK Pathway in 3T3-L1 Adipocytes

**DOI:** 10.3390/nu15030503

**Published:** 2023-01-18

**Authors:** Tomohiko Nakao, Shiro Otaki, Yuri Kominami, Soichi Watanabe, Miho Ito, Teruki Aizawa, Yusuke Akahori, Hideki Ushio

**Affiliations:** 1Department of Aquatic Bioscience, Graduate School of Agricultural and Life Sciences, University of Tokyo, Tokyo 113-8657, Japan; 2Yaizu Suisankagaku Industry Co., Ltd., 5-8-13 Kogawa-shimmachi, Yaizu, Shizuoka 425-8570, Japan

**Keywords:** L-fucose, anti-obesity, AMPK, lipolysis, insulin signaling

## Abstract

L-fucose (Fuc), a monosaccharide with different biological functions in various organisms, exhibits potent anti-obesity effects in obese mice. However, the mechanisms underlying its anti-obesity effects remain largely unknown. In this study, we aimed to investigate the effects of Fuc on lipid metabolism and insulin signaling in 3T3-L1 adipocytes. We found that Fuc treatment suppressed lipid accumulation during adipocyte differentiation. Additionally, Fuc treatment enhanced the phosphorylation of AMP-activated kinase (AMPK) and its downstream pathways, responsible for the regulation of fatty acid oxidation and lipolysis. Furthermore, Fuc-induced activation of the AMPK pathway was diminished by the AMPK inhibitor Compound C, and Fuc treatment considerably promoted glucose uptake via Akt activation in an insulin-resistant state. These findings provide a basis for elucidating the mechanism underlying the anti-obesity effect of Fuc, which may, in the future, be considered as a therapeutic compound for treating obesity and related diseases.

## 1. Introduction

Obesity is caused by an imbalance between energy intake and expenditure and is a serious health problem worldwide. The World Health Organization (WHO) classifies obesity according to body mass index (BMI), which is obtained by dividing the weight in kilograms by the square of the height meters (kg/m^2^). The BMI of 18.5–24.9 kg/m^2^ is normal, 25–29.9 kg/m^2^ is overweight, and >30 kg/m^2^ indicates obesity. The global obese population has tripled from 1975 to 2016. The population of overweight individuals was more than 1.9 billion, and over 650 million adults and over 340 million children constituted the obese population [1]. The increase in the obese population is expected to continue, and it is estimated that by 2030, 38% of adults worldwide will be overweight, and 20% will be obese [2]. Obesity is closely associated with the increasing prevalence of various chronic metabolic disorders, such as type 2 diabetes, hypertonia, and dyslipidemia [3]. Therefore, an increase in the obese population is expected to increase the number of people suffering from obesity-caused lifestyle-related diseases; thus, the prevention of obesity is urgently needed. Moreover, obesity is not only associated with chronic metabolic disorders but is also known to affect climate change issues in recent years. The primary cause of global warming is excessive emissions of greenhouse gas (GHG) from natural systems and human activities. Human activities, including the combustion of fossil fuels, food production, and industrial processes have been considered the source of excessive GHG emissions. However, Magkos et al. have reported that the GHG emissions of an obese individual could be 20% higher compared with the normal weight state according to estimation based on metabolism, food production, and transportation, and the total impact of obesity on a global scale may be extra emissions of ~700 megatons per year of CO2eq, which is about 1.6% of worldwide GHG emissions [4]. Therefore, an increase in the obese population is also an important contributor to global GHG emissions, and both the prevention and treatment of obesity will not only help public health problems but will also contribute to the reduction of GHG emissions.

Adipose tissue is an important energy storage that stores excess energy as triglycerides and resupplies it throughout the body according to the nutritional status of the organism. Adipose tissue maintains normal body weight by regulating a balance between anabolism and catabolism, but an imbalance in this balance can induce obesity. Moreover, adipose tissue not only plays the role of energy storage organs as previously thought but also functions as endocrine organs that secrete various physiologically active substances collectively called adipocytokines. It is suggested that the dysregulation of adipocytokines is closely associated with the development of insulin resistance, type 2 diabetes, and cardiovascular disease, which have attracted considerable attention [5]. Therefore, the homeostasis of adipose tissue is important in the prevention of obesity.

AMP-activated protein kinase (AMPK) plays a key role as a metabolic energy sensor in the regulation of cellular energy homeostasis. AMPK activation improves lipid metabolism by suppressing ATP-consuming synthetic pathways, such as lipogenesis and promoting fatty acid oxidation. AMPK is therefore a useful target for drugs and natural therapeutics in treating metabolic diseases. AMPK activity is regulated in response to stresses that elevate the cellular AMP/ATP ratio, such as hypoglycemia, hypoxia, ischemia, and heat shock. In addition to allosteric activation by AMP, AMPK is activated through the phosphorylation of a critically self-contained threonine residue (Thr172) by upstream kinases [6]. AMPK phosphorylation promotes substrate phosphorylation and deactivates acetyl-CoA carboxylase (ACC), which produces malonyl-CoA, the substrate necessary for fatty acid biosynthesis, thereby inhibiting fatty acid synthesis and simultaneously promoting fatty acid oxidation [7]. AMPK also regulates lipolysis by activating rate-limiting enzymes such as hormone-sensitive lipase (HSL) and adipose triglyceride lipase (ATGL) [8,9]. ATGL catalyzes the first step of triacylglycerol (TAG) hydrolysis, resulting in the formation of diacylglycerol (DAG) and free fatty acids (FFAs) [10]. HSL activation is the second step in the conversion of DAG to monoacylglycerols (MAG) and FFAs [11]. FFAs are used by adipocytes for fatty acid oxidation and can be transported and stored in the peripheral tissues. Peroxisome proliferator-activated receptor γ (PPARγ) is a ligand-activated transcription factor mainly involved in adipocyte differentiation, lipid metabolism, and insulin sensitivity [12]. Previous studies have shown that AMPK activators, such as 5-aminoimidazole-4-carboxamide-1-β-D-ribofuranoside, metformin, and curcumin, suppress adipocyte differentiation by downregulating PPARγ [13,14,15].

The phosphatidylinositol-3 kinase (PI3K)/protein kinase B (Akt) plays crucial roles in insulin-mediated glucose metabolism pathways. Adipocytes express the non-insulin-sensitive glucose transporter 1 (GLUT1) and insulin-sensitive GLUT4, the latter of which is responsible for glucose uptake by adipocytes [16]. Insulin-mediated activation of PI3K/Akt promotes the translocation and fusion of GLUT4 from intracellular vesicles to the plasma membrane, resulting in an increased glucose uptake [17]. As obesity progresses, adipocytes develop hypertrophy, resulting in severe impairment of insulin signaling, which is known to contribute to insulin resistance [18]. Thus, there has been considerable interest in identifying compounds for the treatment of insulin resistance.

Brown seaweed is a common form of edible seaweed in Korea and Japan and is considered a major source of bioactive compounds, such as fatty acids, sterols, polyphenols, and polysaccharides [19]. Fucoidan, a sulfated polysaccharide, has been widely investigated for its anti-obesity properties. Previous studies have shown that fucoidan improves high-fat diet-induced obesity in mice, induces AMPK activation, and inhibits lipid accumulation in 3T3-L1 adipocytes [20,21,22]. However, as fucoidan is a heteropolysaccharide, the monosaccharide composition and sequences drastically affect its functional properties [23], rendering quantitative control of its effects during application challenging. Thus, we focused on L-fucose (Fuc), a structural monosaccharide found in fucoidan, which is suitable for use in functional foods because of its relatively high chemical stability and usability in its purified form. Our previous study showed that Fuc effectively suppressed body weight and liver fat mass gain in high-fat diet (HFD)-induced obese mice [24]. Wu et al. also reported that Fuc exhibits an anti-obesity effect in obese mice by modulating gut microbiota [25]. However, the specific molecular mechanisms underlying the anti-obesity effects of Fuc have not been fully elucidated. Therefore, additional studies are necessary to clarify its regulatory effects on adipocytes.

In this study, we hypothesized that Fuc directly exerts a regulatory effect on lipid metabolism in adipocytes. To evaluate this hypothesis, we first investigated whether Fuc inhibits lipid accumulation in an in vitro 3T3-L1 adipocyte cell model. We then investigated the effects of Fuc on lipid metabolism-related proteins. Finally, we examined whether Fuc affects glucose uptake via the PI3K/Akt pathway.

## 2. Materials and Methods

### 2.1. Chemicals and Reagents

Fucose (#16479, ≥98% purity, Figure 1A) was purchased from Funakoshi (Tokyo, Japan). Fetal bovine serum (FBS) was purchased from Thermo Fisher Scientific (Waltham, MA, USA). Dulbecco’s Modified Eagle’s medium (DMEM), insulin, dexamethasone (DEX), and 3-isobutyl-1-methylxanthine (IBMX) were purchased from Fujifilm Wako (Tokyo, Japan). Protease and phosphatase inhibitor cocktails were purchased from Roche Diagnostics (Basel, Switzerland). The following antibodies were used in this study: AMPKα Rabbit (#2532S), Phospho-AMPKα Rabbit mAb (#2535S), ACC Rabbit Ab (#3662), Phospho-ACC Rabbit Ab (#3661), ATGL Rabbit Ab (#2138S), Phospho-HSL (#4126), PPARγ Rabbit mAb (#2443S), Akt Rabbit mAb (#4691S), and Phospho-Akt Rabbit mAb (#13038S); all of these were purchased from Cell Signaling Technology Japan (Tokyo, Japan). β-Actin mouse rabbit mAb (#010-27841) was purchased from Fujifilm Wako. Alexa Fluor 680 goat anti-rabbit IgG (#A21076) and Alexa Fluor 680 goat anti-mouse IgG (#A28183) were purchased from Thermo Fisher Scientific.

### 2.2. Cell Culture and Adipocyte Differentiation

3T3-L1 murine preadipocytes were purchased from the Japanese Collection of Research Bioresources Cell Bank (JCRB, Osaka, Japan). Cells were maintained in low-glucose DMEM, containing 10% FBS, at 37 °C in 5% CO_2_. The preadipocytes were seeded into 24-well plates at a density of 2 × 10^4^ cells/well and cultured for two days until reaching confluence. After confluence, the medium was replaced and cultured for an additional two days. Differentiation was induced using 5 µg/mL insulin, 1 µM DEX, and 0.5 mM IBMX in DMEM supplemented with 10% FBS. After two days, the cells were transferred to DMEM supplemented with 10% FBS and 5 µg/mL insulin for two days. Subsequently, DMEM supplemented with 10% FBS was renewed every alternate day until the cells were fully differentiated. The differentiated adipocytes were treated with Fuc for the indicated time periods. In the AMPK inhibitor Compound C (CC) experiments, adipocytes were pretreated with 10 µM CC for 1 h before treatment with Fuc and CC.

### 2.3. Cell Viability Assay

Cell viability was analyzed using the MTT assay. The 3T3-L1 preadipocytes were seeded into 96-well plates at a density of 1 × 10^3^ cells/well and incubated for 24 h. Preadipocytes and differentiated adipocytes were treated with various concentrations (0–30 mM) of Fuc for 24 and 48 h. MTT solution was added to each well, and the cells were incubated for 4 h at 37 °C. The medium was removed, formazan was dissolved in DMSO, and the absorbance was detected with a spectrophotometer (SPECTRA MAX, Molecular Devices, CA, USA) at 570 nm.

### 2.4. Oil Red O Staining

The cells were seeded into 12-well plates at a density of 4 × 10^4^ cells/well and differentiated into adipocytes for eight days with or without Fuc. Adipocytes were washed twice with PBS and fixed in 4% paraformaldehyde for 20 min at room temperature. The cells were then stained with an Oil Red O working solution. The stained cells were then washed once with 60% isopropanol and twice with PBS. Images were obtained via microscopy (IX70, Olympus, Tokyo, Japan); the stained oil droplets were dissolved in isopropanol and quantified at 510 nm using a spectrophotometer and standardized by cell numbers counted using Image j software (NIH, Bethesda, MD, USA).

### 2.5. Western Blot Analysis

The cells were lysed in radioimmunoprecipitation assay lysis buffer (50 mM Tris pH 7.5, 150 mM NaCl, 1% nonidet P-40, 0.5% sodium deoxycholate, and 0.1% sodium dodecyl sulfate) containing a protease and phosphatase inhibitor cocktail. Proteins were quantified using a BCA assay kit according to the supplier’s instructions (Thermo Fisher Scientific). Total protein was separated using sodium dodecyl sulfate-poly-acrylamide gel electrophoresis (SDS-PAGE) and transferred onto polyvinylidene difluoride (PVDF) membranes (Sigma-Aldrich, Burlington, MA, USA). After blocking with Intercept^®^ Blocking Buffer (LI-COR, Lincoln, NE, USA) at room temperature for 1 h, the membranes were washed three times with Tris-buffered saline containing 0.05% Tween 20 (TBS-T), and the membranes were incubated with the primary antibody overnight at 4 °C. After washing with TBS-T, the membranes were incubated with a secondary antibody at room temperature for 1 h and washed with TBS-T. Immunoreactive proteins were visualized using the Odyssey Fc Imaging System (LI-COR, NE, USA). Band intensities were quantified using the ImageJ software (NIH).

### 2.6. Glucose Uptake Assay

Glucose uptake by adipocytes was determined using a fluorescent derivative of glucose (2-NBDG; Abcam, Tokyo, Japan). Adipocytes were treated with Fuc for 24 h, followed by incubation in a glucose-free medium, containing 100 µM 2-NBDG and 100 nM insulin, for 30 min. After washing the cells with PBS, fluorescence was measured at a wavelength of 460/540 nm. Lipopolysaccharide (LPS), tumor necrosis factor-α(TNFα), and interferon-γ (IFNγ) (LTI) were used to induce insulin resistance in adipocytes [26]. Adipocytes pre-treated with Fuc for 4 h were treated with 20 mM Fuc, 1 µg/mL LPS, 10 ng/mL TNFα, and 10 ng/mL IFNγ for 20 h followed by 2-NBDG and insulin for 30 min.

### 2.7. Statistical Analysis

All data are presented as the mean ± standard deviation of values from at least three independent experiments. Statistical analyses were performed using R v4.0.4. In cases involving two groups, the data were analyzed using Welch’s *t*-test. Differences between more than three experimental groups were analyzed using Dunnett’s test. *p-*values < 0.05, 0.01, and 0.001 were considered statistically significant.

## 3. Results

### 3.1. Effects of Fuc on Cell Viability and Lipid Accumulation

First, we performed an MTT assay to evaluate whether Fuc affects the proliferation of 3T3-L1 preadipocytes and adipocytes. Preadipocytes and adipocytes were treated with 0–30 mM Fuc for 24 and 48 h. The results showed that Fuc did not affect the viability of either preadipocytes or adipocytes at the concentrations used in this experiment (Figure 1B,C). Thus, we used Fuc concentrations of 1, 5, 10, and 20 mM for further experiments. To examine the effects of Fuc on lipid accumulation during differentiation, 3T3-L1 preadipocytes were differentiated with a differentiation medium and the indicated concentrations of Fuc for eight days. Subsequently, the cells were stained with Oil Red O. As shown by representative images of adipocytes, Fuc treatment reduced lipid accumulation in adipocytes in a dose-dependent manner. In addition, Fuc treatment promoted the development of smaller lipid droplets in lower numbers than those in the control. The most effective concentration for inhibition was 20 mM Fuc (Figure 1D). Treatment with 5, 10, and 20 mM Fuc significantly reduced cellular lipid content by 13.4%, 28.6%, and 35.1%, respectively, compared to that in the control (Figure 1E). These results suggest that Fuc suppresses lipid accumulation during adipocyte differentiation.

### 3.2. Effects of Fuc on Lipid Metabolism-Related Proteins

Next, we investigated the effect of Fuc on lipid metabolism-related proteins. First, the adipocytes were treated with 20 mM Fuc for the indicated times. As shown in Figure 2A, Fuc treatment promoted AMPK phosphorylation for 5 min. The activation of AMPK reached its maximum at 30 min and continued until 24 h compared to that in the control. Next, we examined the protein expression of p-AMPK and its downstream target, p-ACC, to investigate the effects of Fuc on lipid catabolism. Fuc treatment significantly increased the levels of p-AMPK and p-ACC (Figure 2B,C). Furthermore, we examined the protein expression of p-HSL and ATGL to investigate the effects of Fuc on lipolysis. Fuc treatment significantly increased the protein expression of p-HSL and ATGL. Additionally, we examined the protein expression of PPARγ1 and 2 to investigate the effects of Fuc on adipogenesis-related transcription factor proteins. Fuc treatment significantly reduced PPARγ1 expression and increased PPARγ2 expression (Figure 2D,E). These findings indicate that Fuc can upregulate lipid catabolism in adipocytes.

### 3.3. Effects of Fuc and Compound C on Lipid Metabolism-Related Proteins

To confirm whether the effects of Fuc on lipid metabolism are mediated by AMPK activation, we co-treated adipocytes with Fuc and Compound C, an AMPK inhibitor. As shown in Figure 3, Fuc treatment significantly promoted p-AMPK, p-ACC, p-HSL, ATGL, and PPARγ2 expression, while it inhibited PPARγ1 expression; these results were consistent with those shown in Figure 2. On the other hand, co-treatment with Compound C inhibited the effects of Fuc. Interestingly, the decrease in PPARγ1 expression caused by Fuc was unaffected by co-treatment with Compound C, whereas the increase in PPARγ2 expression was abolished.

### 3.4. Glucose Uptake and Akt Phosphorylation in Adipocytes Treated with Fuc and Insulin

The effect of Fuc on glucose uptake in adipocytes was evaluated using the 2-NBDG uptake assay. Fuc treatment significantly increased the uptake of 2-NBDG in the presence of insulin, whereas no significant differences were observed in response to Fuc treatment in the absence of insulin (Figure 4A). We then determined Akt phosphorylation levels to investigate whether Fuc affects insulin signaling. As shown in Figure 4B, Akt phosphorylation was not affected by Fuc in the absence of insulin but was significantly enhanced by Fuc in the presence of insulin. These results indicate that Fuc enhances glucose uptake through Akt activation and improves insulin sensitivity.

### 3.5. Glucose Uptake and Akt Phosphorylation in Insulin-Resistant Adipocytes Treated with Fuc and Insulin

Fuc treatment enhanced insulin-stimulated glucose uptake and Akt phosphorylation. We then examined whether Fuc ameliorates the LTI-induced impairment of insulin signaling and insulin-stimulated glucose uptake. As shown in Figure 5A, insulin treatment significantly increased glucose uptake and Akt phosphorylation compared with those in the negative control. Fuc treatment significantly increased the uptake of 2-NBDG and Akt phosphorylation compared to those with insulin alone. These results are consistent with Figure 4. In contrast, LTI treatment reduced glucose uptake by 51.6% compared to that with insulin alone, which indicated the presence of insulin resistance, while Fuc treatment increased the reduction of glucose uptake to levels comparable to those untreated with LTI. LTI treatment also reduced insulin-mediated Akt activation, whereas Fuc treatment increased the reduction of Akt activation to levels comparable to those untreated with LTI (Figure 5B). These results indicate that Fuc ameliorates the insulin signaling impairment caused by insulin resistance.

## 4. Discussion

To elucidate the effects of Fuc on lipid metabolism and insulin signaling in adipocytes, we used 3T3-L1 adipocytes to demonstrate that Fuc suppresses intracellular lipid accumulation in adipocytes. We found that Fuc activated the AMPK signaling pathway in adipocytes, suggesting the promotion of lipid catabolism. We also observed that Fuc enhanced the insulin-stimulated glucose uptake via Akt activation and reversed the impairment of insulin signaling by insulin resistance. This is the first study to show that Fuc induces activation of the AMPK and PI3K/Akt pathways, which may mediate the inhibitory mechanism of Fuc against lipid accumulation and insulin resistance.

The AMPK pathway plays a crucial role in lipid metabolism by promoting catabolism and suppressing anabolic processes. AMPK inactivates its substrate, ACC, resulting in the promotion of fatty acid oxidation and the suppression of fatty acid synthesis. In obesity, excessive energy intake leads to increased lipid storage in adipose tissue and reduces AMPK activity [27]; thus, activated ACC promotes fatty acid synthesis. In the present study, treatment with >10 mM Fuc concentrations enhanced the phosphorylation of AMPK and ACC, suggesting that the activation of the AMPK pathway inhibits lipid accumulation. On the other hand, Oil Red O staining results showed that >5 mM Fuc concentrations inhibited lipid accumulation, which is inconsistent with reports regarding its effects on AMPK and ACC; some studies have shown that mannose activates AMPK in Saos-2 cells and suppresses adipocyte differentiation by inhibiting the mitogen-activated protein kinase (MAPK) pathway [28,29]. Thus, Fuc may reduce lipid accumulation via different pathways.

It has been reported that berberine and Ginkgolide C activate AMPK and promote lipolysis by upregulating HSL and ATGL and that their effects are abolished by the AMPK inhibitor CC [30,31]. The present results are consistent with those reported previously, suggesting that Fuc promotes lipolysis through AMPK activation.

PPARγ has two main isoforms, PPARγ1 and γ2, which are transcribed from the same gene and regulate different genes in adipocytes [32]. AMPK inhibits adipocyte differentiation by reducing PPARγ expression [13,14,15]. Therefore, while we expected that Fuc treatment would reduce the protein expression of both PPARγ1 and γ2, our results showed that PPARγ2 protein expression increased, while PPARγ1 expression decreased. Furthermore, the Fuc-mediated decrease in PPARγ1 expression was not affected by the AMPK inhibitor CC, whereas the increase in PPARγ2 expression was abolished, suggesting that these effects are mediated by separate pathways. It has been reported that PPARγ2 improves insulin sensitivity [33,34]. Its distinct roles, however, are still controversial, although functional differences between PPARγ1 and γ2 have been investigated.

It is known that caloric restriction triggers the activation of AMPK; therefore, Calorie Restriction Mimetics (CRMs), compounds exhibiting calorie restriction effects, are considered important anti-obesity targets. Monosaccharides such as D-glucosamine, 2-deoxy-glucose, and D-allulose have been identified as CRMs and activate AMPK by entering cells through GLUTs and inhibiting glycolysis [35]. We observed the influx of Fuc in adipocytes treated with Fuc using GC-MS/MS, which suggested that Fuc is also a monosaccharide that may function as a CRM by inhibiting glycolysis. the Further studies are necessary to determine whether Fuc activates AMPK as a CRM by inhibiting glycolysis [36].

Recent reports have suggested that chronic inflammation is closely associated with insulin resistance [37]. Adipocytes exhibiting progressive hypertrophy secrete proinflammatory cytokines and promote chronic inflammation in adipose tissue through autocrine and paracrine mechanisms, resulting in insulin resistance. Insulin resistance results in reduced glucose uptake by impairing insulin signaling, which is consistent with our results; that is, LPS, TNFα, and IFNγ (LTI) treatment markedly suppresses glucose uptake and Akt activation. Dobashi et al. [26] have suggested a useful method for inducing insulin resistance by treating adipocytes with LTI. This study successfully demonstrated that Fuc enhances glucose uptake and Akt activation, even under insulin resistance. AMPK inhibits a key pathway of proinflammatory cytokines, the nuclear factor-κB (NF-κB) pathway [38], suggesting that Fuc may display potent anti-inflammatory effects. However, further studies are necessary to explore the effect of Fuc on the secretion of proinflammatory cytokines and on the NF-κB pathway.

Fuc is a monosaccharide composed of the polysaccharide fucoidan, which can easily be extracted from brown algae [39]. A significant amount of seaweed waste is generated during production, with no effective utilization [40]. Researchers have demonstrated that it is safe to use Fuc as a food additive [41]. In addition, we have reported that Fuc suppressed visceral fat in the human administration test [42], and we consider there are no dosage-related problems in human applications. Our study demonstrated that Fuc exhibits anti-obesity and anti-insulin resistance effects by regulating the AMPK and PI3K/Akt pathways. Therefore, Fuc may represent a promising food additive for the prevention of obesity-induced diseases.

## Figures and Tables

**Figure 1 nutrients-15-00503-f001:**
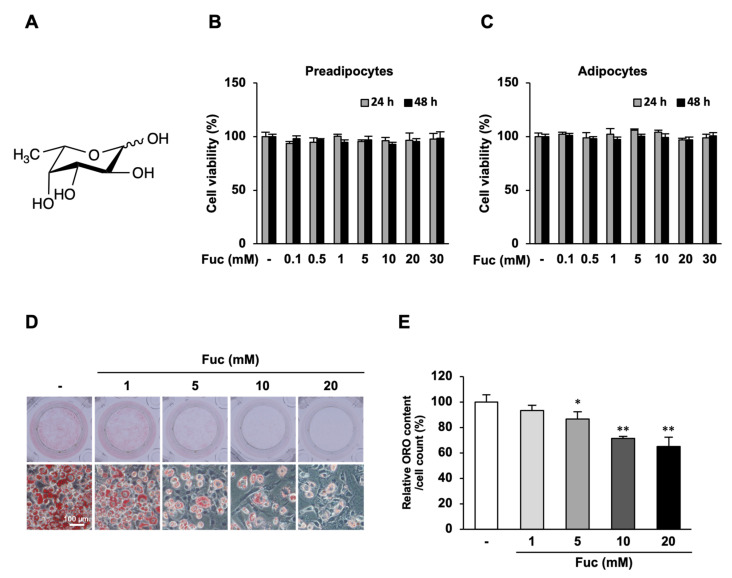
Effects of L-fucose (Fuc) on lipid accumulation during adipocyte differentiation. The cells were differentiated into adipocytes for eight days with or without Fuc and stained with Oil Red O. (**A**) Chemical structure of Fuc. (**B**,**C**) Effects of Fuc on the viability of 3T3-L1 preadipocytes and adipocytes. The cells were treated with the indicated concentrations (0–30 mM) of Fuc for 24 and 48 h. Cell viability was determined using the MTT assay. (**D**) Representative images of Oil Red O-stained adipocytes at 20× magnification (scale bar = 100 µm). (**E**) Quantified Oil Red O staining after isopropanol extraction. Data are presented as mean ± SD (*n* = 6: MTT assay, *n* = 3: Oil Red O staining), and statistical significance was calculated using Dunnett’s test (the group treated without Fuc vs. the groups treated with Fuc, * *p* < 0.05; ** *p* < 0.001).

**Figure 2 nutrients-15-00503-f002:**
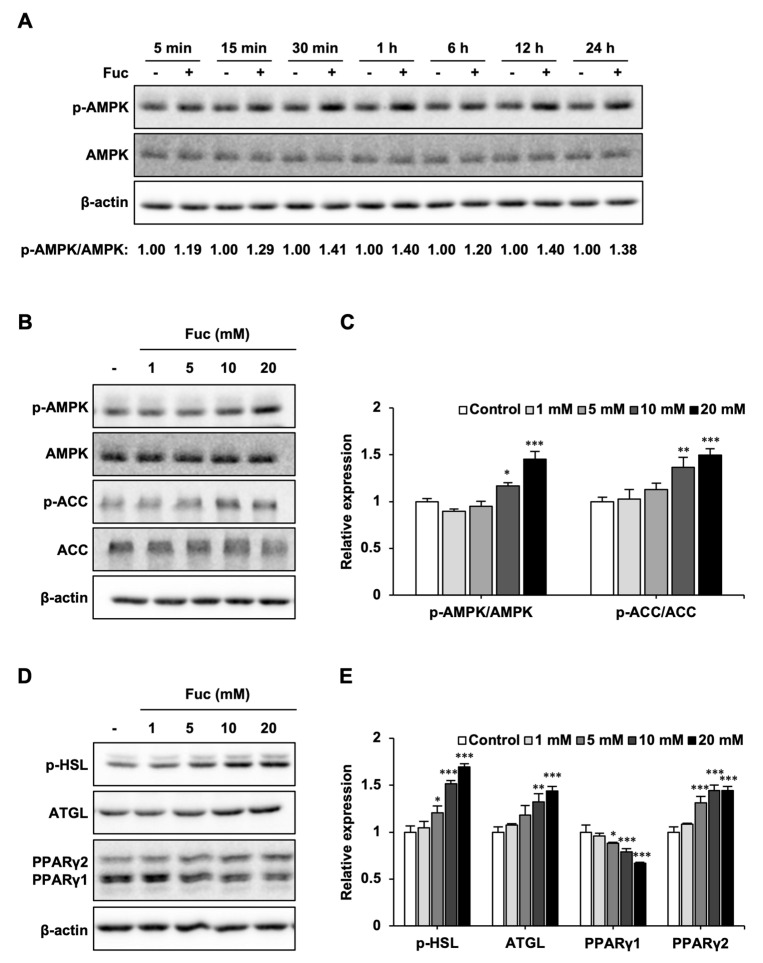
Effects of L-fucose (Fuc) on lipid metabolism-related proteins. (**A**) Time course of AMP-activated protein kinase (AMPK) phosphorylation in adipocytes treated with 20 mM Fuc. (**B**,**C**) Protein expression levels of p-AMPK, AMPK, p-acetyl-CoA carboxylase (ACC), and ACC in adipocytes treated with Fuc for 24 h. (**D**,**E**) Protein expression levels of p-hormone-sensitive lipase (HSL), adipose triglyceride lipase (ATGL), and peroxisome proliferator-activated receptor (PPAR)γ1, 2 in adipocytes treated with Fuc for 24 h. Data are presented as mean ± SD (*n* = 3), and statistical significance was calculated using Dunnett’s test (the group treated without Fuc vs. the groups treated with Fuc, * *p* < 0.05; ** *p* < 0.01; *** *p* < 0.001).

**Figure 3 nutrients-15-00503-f003:**
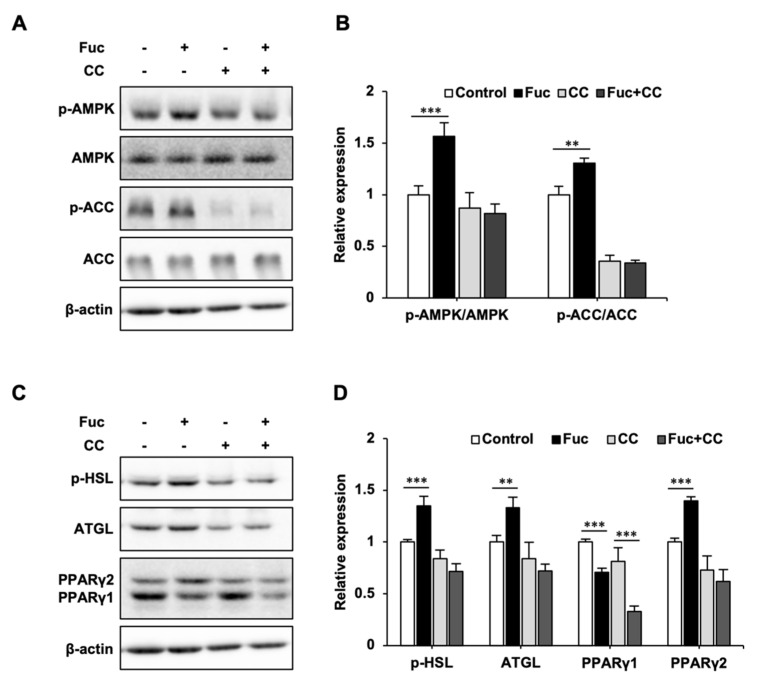
Effects of L-fucose (Fuc) and AMP-activated protein kinase (AMPK) inhibitor Compound C (CC) on lipid metabolism-related proteins. (**A**,**B**) Protein expression levels of p-AMPK, AMPK, p-acetyl-CoA carboxylase (ACC), and ACC in adipocytes cotreated with 10 µM CC and Fuc for 24 h. (**C**,**D**) Protein expression levels of p-hormone-sensitive lipase (HSL), adipose triglyceride lipase (ATGL), and peroxisome proliferator-activated receptor (PPAR)γ1, 2 in adipocytes cotreated with 10 µM CC and Fuc for 24 h. Data are presented as mean ± SD (*n* = 3), and statistical significance was calculated using Welch’s *t*-test (control vs. Fuc, CC vs. Fuc + CC, ** *p* < 0.01; *** *p* < 0.001).

**Figure 4 nutrients-15-00503-f004:**
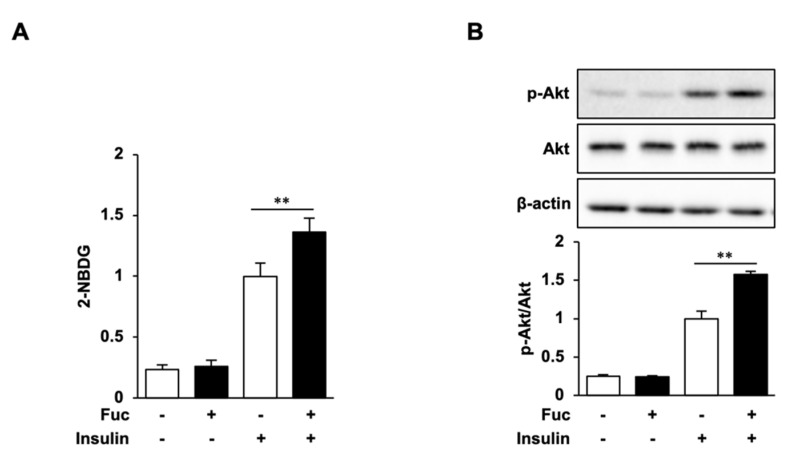
Effects of L-fucose (Fuc) on glucose uptake and Akt phosphorylation. (**A**,**B**) Adipocytes treated with Fuc for 24 h were stimulated with insulin for 30 min. To estimate the glucose uptake, adipocytes were stimulated with 2-NBDG and insulin for 30 min. Data are presented as mean ± SD (*n* = 3), and statistical significance was calculated using Welch’s *t*-test (control vs. Fuc, CC vs. Fuc + CC, ** *p* < 0.01).

**Figure 5 nutrients-15-00503-f005:**
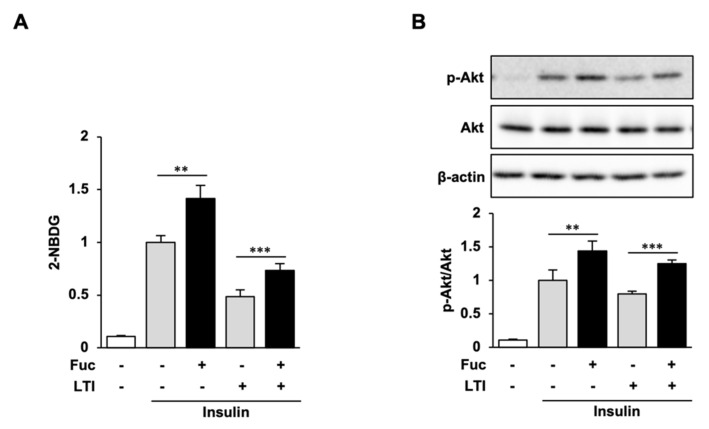
Effects of L-fucose (Fuc) on glucose uptake and Akt phosphorylation in insulin-resistant adipocytes. (**A**,**B**) Adipocytes pretreated with Fuc for 4 h were exposed to a combination of Fuc and LPS, TNFα, and IFNγ (LTI) for 20 h. To estimate the glucose uptake, adipocytes treated with Fuc and LTI were stimulated with 2-NBDG and insulin for 30 min. Data are presented as mean ± SD, (*n* = 3) and statistical significance was calculated using Welch’s *t*-test (control vs. Fuc, CC vs. Fuc + CC, ** *p* < 0.01; *** *p* < 0.001).

## Data Availability

The data presented in this study are available from the corresponding author upon reasonable request.

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
