# Peer review of "L-Fucose Suppresses Lipid Accumulation via the AMPK Pathway in 3T3-L1 Adipocytes"

_nutrients, 2023, doi:10.3390/nu15030503_

Round 1

Reviewer 1 Report

    This manuscript reported an effect of L-fucose (Fuc) on lipid metabolism and insulin signaling in 3T3-L1 cells. This study would have some scientific merits; however, there are some critical issues which have to be properly addressed.

1.     In 2.3 cell Viability assay and the related result sections, the cell density of 3T3-L1 cells plated in a 96-well plate was not described.

2.     There are no data showing that Fuc treatment does not affect the proliferation of 3T3-L1 preadipocytes and adipocytes (lines 150-154) because Figures 1B and 1C only displayed their viability at a single time point. In Figure 1D, the number of adipocytes seems to be different depending on Fuc concentrations at the Oil Red O staining. Therefore, the total number of 3T3-E1 adipocytes at the end of  the differentiation should be assessed or the quantitative data of Figure 1E should be expressed after adjusting the values per g cell protein.

3.     The quantitative data of Figure 2A should be added to make it clearer the authors’ point described in lines 175-177.

4.     In the discrepancy of PPARg1 and g2 in Figure 2E, are there any data in the different treatment  time points other than 24h or mRNA level evaluation?

5.     The description in lines 231-234 is ambiguous; (1) are the authors only comparing between insulin + and - in cells treated Fuc-/LTI- cells here? The effect of Fuc in LTI- insulin+ cells should be mentioned as well.  (2) the words “Fuc treatment restored glucose uptake or Akt activation” do not seem to be the best suitable. Fuc enhances an effect of insulin on glucose uptake and p-Akt/Akt by about 30% and LTI treatment decreased them similarly in both cells treated with and  without Fuc by about 50%. Thus, the summary glucose uptake and p-Akt/Akt levels in the cells treated with Fuc, insulin, and LTI apparently settled to the similar levels to those of the control cells  treated insulin.

6.     Please consider a revision of a sentence in lines 241-242.

Author Response

We appreciate the reviewer 1 very much or the constructive and valuable comments.

  1. In 2.3 cell Viability assay and the related result sections, the cell density of 3T3-L1 cells plated in a 96-well plate was not described.

We added the description of cell density plated in 96-well plates (lines 108-110).

  1. There are no data showing that Fuc treatment does not affect the proliferation of 3T3-L1 preadipocytes and adipocytes (lines 150-154) because Figures 1B and 1C only displayed their viability at a single time point. In Figure 1D, the number of adipocytes seems to be different depending on Fuc concentrations at the Oil Red O staining. Therefore, the total number of 3T3-E1 adipocytes at the end of  the differentiation should be assessed or the quantitative data of Figure 1E should be expressed after adjusting the values per g cell protein.

In addition to the 48 h MTT assay results, 24 h results were also shown (Figure 1B and 1C). Since no significant differences were found at either time point, these data show that Fuc (1-30 mM) does not affect cell viability.

In Figure 1E, corrections were conducted based on the cell count using microscopic images. Statistical analysis was performed on the newly obtained values.

  1. The quantitative data of Figure 2A should be added to make it clearer the authors’ point described in lines 175-177.

In some reports, parts of the results have shown the time course of p-AMPK with N=1 only (Igata et al., 2005; Zhou et al., 2009), so we designed our experiments based on these. Since the experiment was conducted with N=1 only, we added the notation of quantitative values with the control group at each time point as 1.

  1. In the discrepancy of PPARg1 and g2 in Figure 2E, are there any data in the different treatment  time points other than 24h or mRNA level evaluation?

Total mRNA for PPARg was reduced by Fuc treatment. The same results were obtained in experiments with cells treated with Fuc for 8 days after adipocyte differentiation. Besides, metformin was used as a positive control in the same experiment, and metformin reduced both PPARg1 and 2. Therefore, we consider this effect to be fucose-specific. We will investigate to understand the mechanisms for the fucose-specific effects and publish the details elsewhere.

  1. The description in lines 231-234 is ambiguous; (1) are the authors only comparing between insulin + and - in cells treated Fuc-/LTI- cells here? The effect of Fuc in LTI- insulin+ cells should be mentioned as well.  (2) the words “Fuc treatment restored glucose uptake or Akt activation” do not seem to be the best suitable. Fuc enhances an effect of insulin on glucose uptake and p-Akt/Akt by about 30% and LTI treatment decreased them similarly in both cells treated with and  without Fuc by about 50%. Thus, the summary glucose uptake and p-Akt/Akt levels in the cells treated with Fuc, insulin, and LTI apparently settled to the similar levels to those of the control cells  treated insulin.

(1) As a replication of Figure 4, we compared insulin- vs. insulin+, and Fuc- vs. Fuc+ in the presence of insulin. We added the description of the effect of Fuc on LTI- insulin+ and showed results consistent with Figure 4 (lines 243-246).

(2) I have confused you in terms of "restore". As you mentioned, I rewrote "Fuc increased the reduction of glucose uptake and Akt activation by LTI treatment to levels comparable to those untreated with LTI" (lines 248-250).

  1. Please consider a revision of a sentence in lines 241-242.

We have changed the text.

(Before) For the glucose uptake experiments

(After) To estimate the glucose uptake (lines 261)

Reviewer 2 Report

In this manuscript, Nakao et al., showed that L-Fucose (Fuc) suppressed lipid accumulation during adipocyte differentiation through increasing the phosphorylation of AMP-activated kinase (AMPK). The authors also showed that Fuc treatment promoted glucose uptake via Akt activation. The current manuscript is interesting as they showed the effects of fucose in adipocytes. However, this reviewer asks some modifications before accepting the manuscript for publication. Specific comments are shown below.  

1.     In line 16, the authors wrote “AMP-activated kinase (AMPK) and its downstream pathways, including fatty acid oxidation and lipolysis”. Although the authors provide the evidence for p-HSL and ATGL levels by fucose treatment, effects of fucose on fatty acid oxidation and lipolysis were not directly supported by the data. Effects on fatty acid oxidation and lipolysis would be revised by additional data or should be deleted. 

2.     The authors data showed that fucose stimulated AMPK and AKT activities. Which kinase acts as an upstream factor on the other kinase?

3.     In fig 2A, p-AMPK/AMPK can be quantified similar to that in Fig 2C.

4.     The authors used purified Fucose from Funakoshi. The purity and/or HPLC analytic data would be nice to be included in the paper.   

5.     Fucose increased p-HSL, p-ACC, p-AMPK, and Pparg2 levels. Although Pparg1 expresion was reduced, increased Pparg2 is associated with higher adipocyte differentiation.  As Fucose inhibited adipocyte differentiation and increased Pparg2 expression, how do the authors reconcile these somewhat contradictory effects? Also, can the authors verify the pparg2 and pparg1 proteins in western blots? Another way to detect or verify the pparg2 and pparg1 may be necessary.

6.     The authors showed that 10-20mM fucose inhibited adipocyte differentiation and glucose uptake. Can this amount of fucose be potentially translated into human application? The potential problems for Human application should be discussed.     

Author Response

The authors appreciate the reviewer 2 for the valuable and constructive comments.

  1. In line 16, the authors wrote “AMP-activated kinase (AMPK) and its downstream pathways, including fatty acid oxidation and lipolysis”. Although the authors provide the evidence for p-HSL and ATGL levels by fucose treatment, effects of fucose on fatty acid oxidation and lipolysis were not directly supported by the data. Effects on fatty acid oxidation and lipolysis would be revised by additional data or should be deleted.
    Our results showed that Fuc enhanced p-AMPK, p-ACC, p-HSL, and ATGL. Numerous studies have reported that ACC inactivation by phosphorylation is necessary for promoting fatty acid oxidation. In addition, it is known that activation of HSL and ATGL leads to promoting lipolysis. Therefore, we concluded that Fuc promotes fatty acid oxidation and lipolysis by regulating each key step. However, due to misleading wording, the sentence was changed as follows; Fuc treatment enhanced the phosphorylation of AMP-activated kinase (AMPK) and its downstream pathways, responsible for the regulation of fatty acid oxidation and lipolysis.
  2. The authors data showed that fucose stimulated AMPK and AKT activities. Which kinase acts as an upstream factor on the other kinase?
    The AMPK activators AICAR and metformin activate insulin-mediated Akt, thus AMPK acts as an upstream factor of Akt.
  3. In fig 2A, p-AMPK/AMPK can be quantified similar to that in Fig 2C.
    In some reports, parts of the results have shown the time course of p-AMPK with N=1 only (Igata et al., 2005; Zhou et al., 2009), so we designed our experiments based on these. Since the experiment was conducted with N=1 only, we added the notation of quantitative values with the control group at each time point as 1.
  4. The authors used purified Fucose from Funakoshi. The purity and/or HPLC analytic data would be nice to be included in the paper.
    We added the product number and the purity data for easy access to reagent information (lines 83).
  5. Fucose increased p-HSL, p-ACC, p-AMPK, and Pparg2 levels. Although Pparg1 expresion was reduced, increased Pparg2 is associated with higher adipocyte differentiation. As Fucose inhibited adipocyte differentiation and increased Pparg2 expression, how do the authors reconcile these somewhat contradictory effects? Also, can the authors verify the pparg2 and pparg1 proteins in western blots? Another way to detect or verify the pparg2 and pparg1 may be necessary.
    Total mRNA for PPARg was reduced by Fuc treatment. The same results were obtained in experiments with cells treated with Fuc for 8 days after adipocyte differentiation. Besides, metformin was used as a positive control in the same experiment, and metformin reduced both PPARg1 and 2. Therefore, we consider this effect to be fucose-specific.

If fucose acts as a CRM as described in the statement, it is assumed that fucose can inhibit glycolysis. It is reported that the balance of pparg1 and 2 is altered by nutritional status, and in adipose tissue, PPARg2 is reduced by fasting. Since the results differ from our results and are possibly regulated by a mechanism that is distinct from the AMPK pathway, no conclusions could be drawn. Therefore, we plan to explore the phenotypic differences shown by fucose administration under PPARg knockdown and will show the mechanisms elsewhere.

  1. The authors showed that 10-20mM fucose inhibited adipocyte differentiation and glucose uptake. Can this amount of fucose be potentially translated into human application? The potential problems for Human application should be discussed.
    We have reported that Fuc suppressed visceral fat in the human administration test under 150 mg/day dosage for 20 weeks (Aizawa et al., Effect of l-fucose-containing diet on visceral fat and bowel movement, and its safety: A randomized, double-blind, placebo-controlled, parallel-group study. Pharmacometrics 2021, 100, 63-70). In vivo, relatively small doses may have an anti-obesity effect, thus We do not believe there are any dosage-related problems in human applications and added the statements in the text (lines 329-331).

Round 2

Reviewer 1 Report

All raised concerns were adequately explained and resolved by the authors' responses.

Reviewer 2 Report

Thank you for revising the manuscript.